# Zircon Dating, Geochemistry, and Metallogenic Significance of Early Paleozoic Mafic Rocks in Northern Guangxi Province, China

**Zhuolin Xie** [1,2,*], **Shehong Li** [1,2], **Yu Dai** [3], **Chongjin Pang** [1,2], **Saisai Li** [1,2], **Xuhan Hu** [1,2] and **Jinming Wu** [1,2]

[1] College of Earth Sciences, Guilin University of Technology, Guilin 541004, China; shehongli@glut.edu.cn (S.L.); chongjinpang@glut.edu.cn (C.P.); lanqi178@glut.edu.cn (S.L.); woaixuhan0213@163.com (X.H.); wjmwhatever777@163.com (J.W.)
[2] Guangxi Key Laboratory of Hidden Metallic Ore Deposit Exploration, Guilin 541004, China
[3] Guangxi Institute of Regional Geological Survey, Guilin 541004, China; yudai@glut.edu.cn.com
[*] Correspondence: xiezhuolin1997@163.com

**Abstract:** Magmatic rocks, deformed structures, and tin–polymetallic deposits are widely distributed in the western Jiangnan Orogenic Belt (JOB) of northern Guangxi Province, China. Magmatism and mineralization are believed to have occurred in the Neoproterozoic stage. Herein, we report the zircon U–Pb dating results of investigations on the Ping'an Pb–Zn–Cu polymetallic veins hosted in gabbro near Ping'an Village. Zircon U–Pb dating of the host gabbro yielded ages of $853.0 \pm 7.8$, $837.7 \pm 7.2$, and $450.4 \pm 6.7$ Ma. The younger age represents the emplacement of gabbros, whereas the older Neoproterozoic age reflects inherited zircons from the wall rocks or source regions. The formation of gabbros (Caledonian) is related to the subduction and collisions of microplates in the western JOB, which are controlled by movements of the Yangtze and Cathaysian plates. We consider that the late Caledonian regional shearing in the western JOB resulted in the fracturing and faulting of rocks (Neoproterozoic and early Caledonian), which provided conduits for the flow of hydrothermal fluids and accommodation for the associated mineralization. Geochemistry investigations show that the Caledonian basic magmatic activity provided a certain material source for the final mineralization. We propose that the tin–polymetallic deposits in the northern Guangx Province, and Neoproterozoic cassiterite crystallization, were subjected to Caledonian shear crushing and hydrothermal transformation with copper, lead, zinc, and other metal elements based on our comprehensive analysis, providing a new ideology for understanding the geology and mineralization in this area.

**Keywords:** Caledonian; China; gabbro; Ping'an deposit; Pb–Zn–Cu polymetallic mineralization; Zircon U–Pb dating

## 1. Introduction

The western Jiangnan Orogenic Belt (JOB) is located between the Yangtze and Cathaysian plates in the northern Guangxi Province, China, and contains Proterozoic magmatic rocks and tin–polymetallic deposits [1–7]. The granitic and mafic–ultramafic rocks in this belt are related to the assembly of the Neoproterozoic Rodinia supercontinent [7–10]. However, recent studies have suggested that mafic magmatism in northern Guangxi intruded during the Caledonian orogeny, and zircons have U–Pb age between 570 and 400 Ma and magmatic characteristics in the mafic rocks [11,12]. However, few Caledonian magmatic bodies have been found in the western JOB.

Sedimentary rocks in the JOB area have undergone shear deformation and hydrothermal alteration during the Caledonian orogeny [8,13,14]. Cassiterite U–Pb dating has shown that Sn polymetallic deposits in the JOB area were formed between 834 and 829 Ma, which is consistent with the ages of felsic magmatic rocks, such as the Sanfang, Yuanbaoshan, and

Pingying granitic plutons. Shearing occurred around 419–319 Ma, much later than Sn mineralization, indicating that no genetic relationship exists between the two events [2,15–18]. Many polymetallic deposits are distributed around granites; however, few deposits appeared whose formation was related to magmatic or hydrothermal processes occurring after the Proterozoic period (Figure 1a).

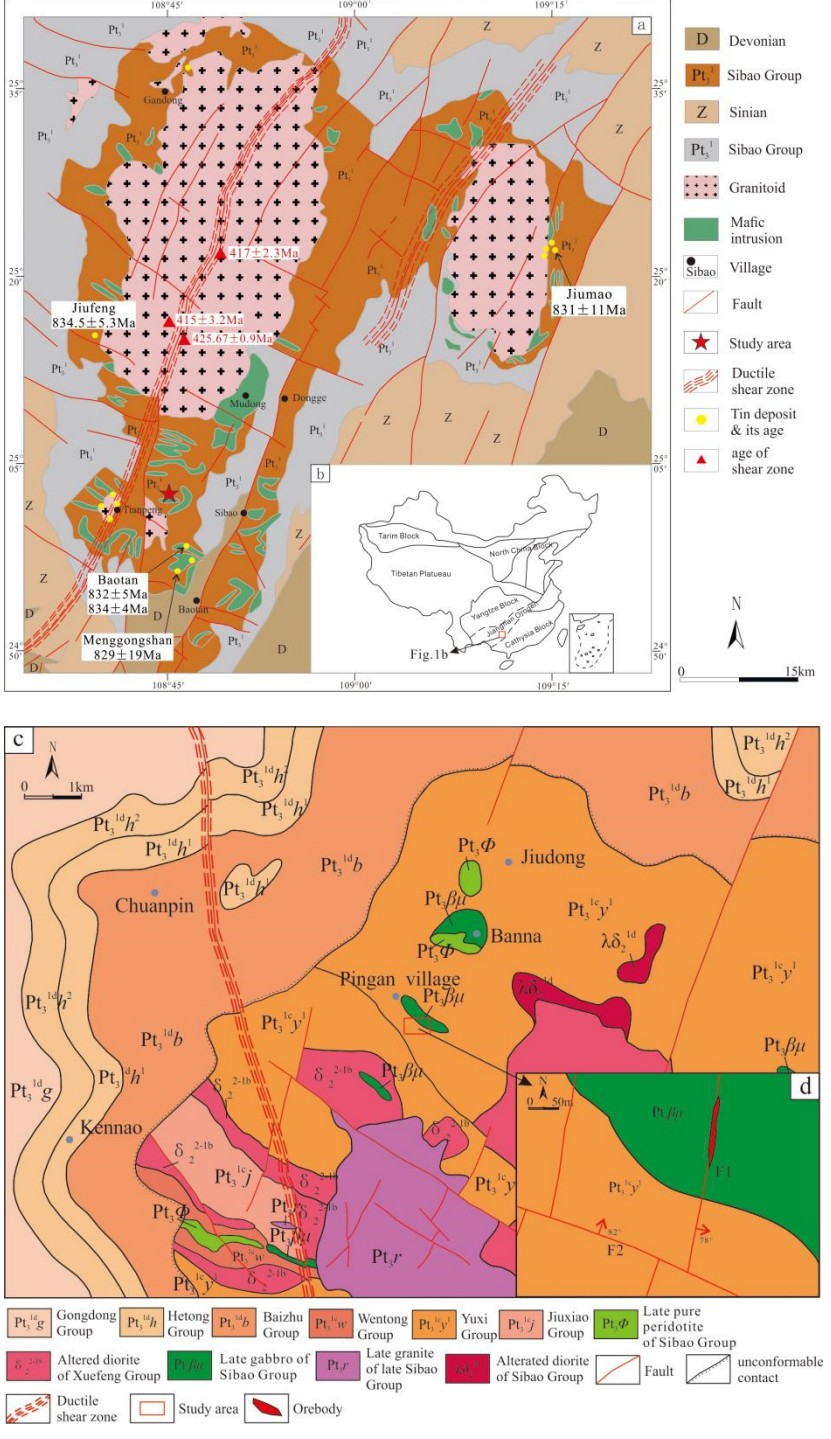

**Figure 1.** Regional geological and mineralization geological maps [1–4,19] (modified after Mao et al., 1988; Wang et al., 2006; Zhang et al., 2016, 2019). (**a**) Simplified geological map of the study region of Jiuwangdashan–Yuanbaoshan, northern Guangxi Province; (**b**) simplified tectonic map of China; (**c**) geological mining map of the Ping'an Pb–Zn–Cu deposit, showing the location of study area; (**d**) geological map of the study area, showing one of the distribution of ore bodies.

Tin deposits in the northern Guangxi Province are characterized by three geological features. First, some ore bodies that have reached industrial grade cannot be used for industry because shearing has reduced the cassiterite particles to sizes <20 μm, and cassiterite minerals in most tin ores are generally fragmented, whereas other associated metal minerals, such as chalcopyrite, sphalerite, and galena, are not. Second, tin mineralization primarily occurs in a large number of alteration zones rather than in the cassiterite minerals or sulfides, and tin is carried into the mineral grain fissure by fine particles. Third, tin polymetallic–ore bodies are distributed near shear zones, fault zones, and felsic and mafic intrusions. Ore bodies are distributed along faults and Caledonian ductile shear zones.

The genetic relationship between Pb–Zn–Cu polymetallic deposits and biotite granite in the region has always been controversial. The tin–polymetallic deposits are more extensively exposed in the region than the Cu, Pb, and Zn-associated metal deposits. The tin deposits in the northern Guangxi Province can be divided into two metallogenic series: the Cu, Ni, Co, and Pb deposit series related to the Sibao ultrabasic and mafic intrusive rocks, and the Sn, Cu, Pb, Zn, Sb, W, Au, and U polymetallic deposit series related to the Xuefeng acidic intrusive rocks (mainly biotite, granite, and granodiorite) [1,20,21]. There are many super-large tin–polymetallic deposits, such as those in Jiumao, Baotan, Jiufeng, and Menggongshan, and no less than 100 tin deposits have been discovered in total. Previous studies on the ages of these tin deposits and magmatic rocks have been widely reported, most of which focus on the Neoproterozoic age of 850 Ma [4,12,22–25]. However, few reports on the Caledonian mafic magmatism and metallogenic time of Caledonian Pb–Zn–Cu deposits in the northern Guangxi Province, except the early Paleozoic granites, Neoproterozoic mafic ultrabasic rocks, and granitic intrusive rocks, have been published. Recently, Xiaofeng obtained a Caledonian ultrabasic rock age of $421.9 \pm 7.8$ Ma (for 24 zircons), suggesting that in addition to the hydrothermal activity, there may also have been magmatic activity in this area during the Caledonian period. Due to the coexistence of magmatic and tectonic activities in the study area, the analysis is difficult. Therefore, there are fewer studies on the Pb–Zn–Cu deposits in the Jiuwandashan area in the northern Guangxi Province. The authors of this study discovered that the gabbro in Ping'an village might have a Caledonian origin. This gives rise to the question of whether there were Caledonian magmatic hydrothermal and tectonic thermal events in this area. In addition, is the Caledonian magmatic activity superimposed on the polymetallic deposits in the northern Guangxi Province? This study, therefore, tries to address the aforementioned questions by analyzing the deposits through ICP–MS U–Pb dating combined with mineralogical geology based on previous studies and provides a basis for the period of mineralization in this area.

In this study, we present an integrated study of the geology, structures, and mineralization of the Ping'an intrusive gabbro and associated Pb–Zn–Cu deposits located near Ping'an village in the northern Guangxi Province.

## 2. Geological Setting

### 2.1. Regional Geology

The study region is located in the south of the Sangfang granite deposits in the northern Guangxi Province, China (Figure 1a,b). The major geological units are the Sibao Group, comprising terrigenous clastic sedimentary rocks interlayered with mafic–ultramafic volcanic rocks, phyllite of the Danzhou Group, and Sinian (800–570 Ma) [26] continental, marine, and pyroclastic rocks. Sediments were sourced mainly from eroded magmatic arc rocks.

The two largest granitic plutons in the study region (Figure 1b) are the Sanfang ($1400 \text{ km}^2$) and Yuanbaoshan ($350 \text{ km}^2$) plutons. In addition, several hundred occurrences of granite, diabase, and ultrabasic peridotite masses have been identified. Various dating methods (galena Pb–Pb, cassiterite U–Pb, and zircon U–Pb) have shown that the intrusive rocks are Proterozoic in age [26–31]. The mafic–ultramafic rocks include gabbro, diabase, olivine pyroxenite, and pyroxenite, all of which host Cu–Ni sulfide mineralization [1,32,33]. The zircon U–Pb dating results have shown that these mafic–ultramafic rocks were formed



mainly during 860–740 Ma [27,31,34,35]. Structurally, the study region (Figure 1b) is characterized by complex W-trending folds cut by late faults and N-trending folds associated with late tectonic deformation. The region contains NNE-, NE-, W-, N-, and NW-trending faults, of which the NNE-trending faults extend to hundreds of kilometers and are transect by the NW-trending faults. The faults were formed through compression or extension and showed multiple phases of activity and associated deformation [9]. The NNE-directed shear deformation is common and led to the fracturing, deformation, and alteration of rocks in the study region. According to the literature, deformation occurred during 460–399 Ma [36,37], suggesting that it was associated with the Caledonian orogeny.

Numerous tin–polymetallic deposits are distributed within the felsic magmatic rocks of the study region. These deposits share many similarities, such as ore-bearing strata (Sibao and Danzhou groups), ore-controlling structures (the NNE- and NE-trending faults and NNE-directed shear deformation), mineralization (Sn–Pb–Zn–Cu), and hydrothermal alteration in the zones of highly fractured rocks (Figure 2b–e). Some studies have identified zoning in these deposits, with Sn polymetallic mineralization within 1.5 km of the contact zone between Neoproterozoic granite and host strata showing a gradual transition to Cu–Pb–Zn mineralization within 5 km of the contact zone [9,16].

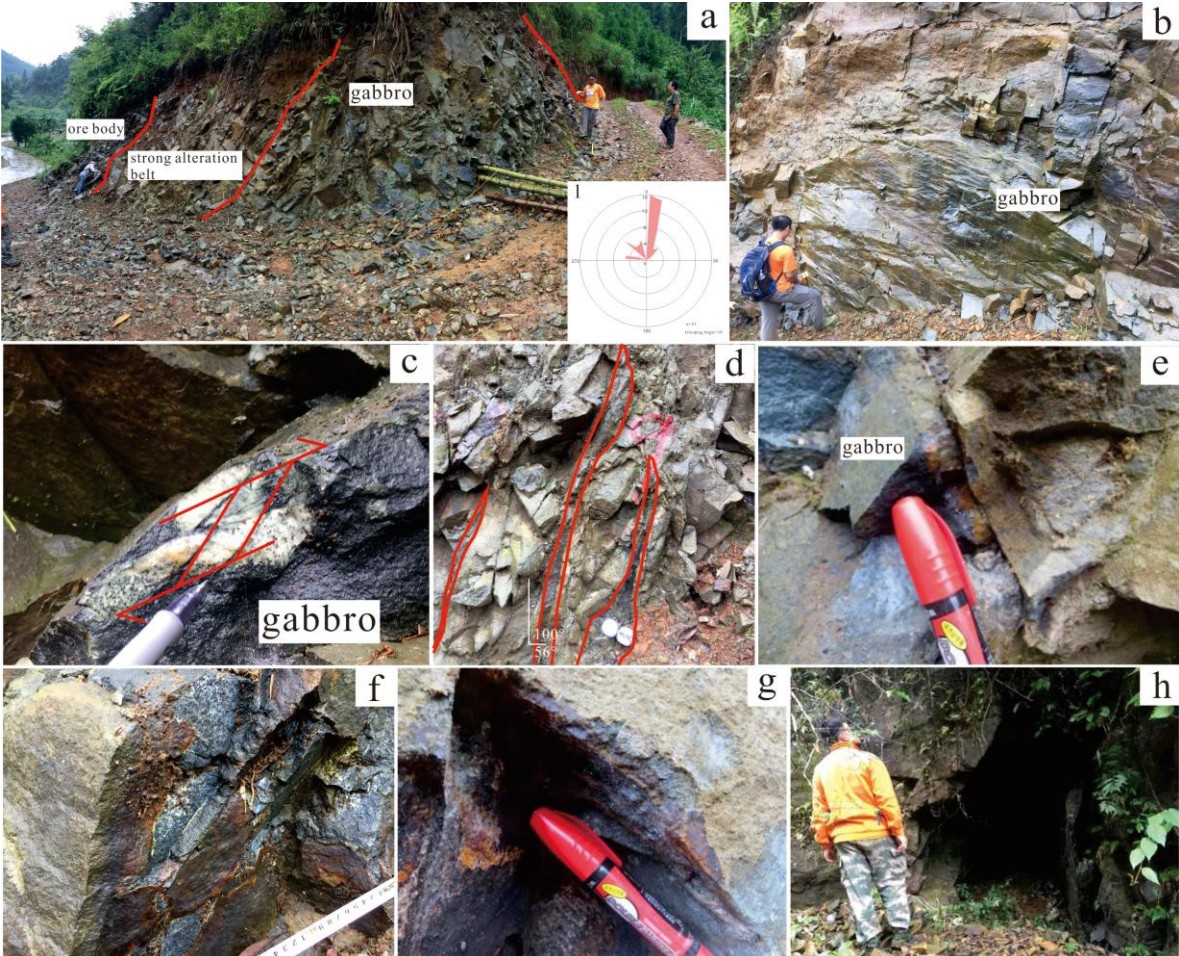

**Figure 2.** Representative outcrop photographs of the Ping'an Pb–Zn–Cu deposit in the northern Guangxi Province. (**a**) Field photograph of gabbro exposure and joint statistics rose diagram; (**b**) field-altered gabbro; (**c**) field photographs of gabbro subjected to brittle–ductile deformation (left-lateral shear); (**d**) traces of hydrothermal entry into the formation; (**e**–**g**) metasomatic alteration mineralization along the fissure; (**h**) mining hole.

### 2.2. Ping'an Area

The Ping'an gabbro is located at the south of Ping'an village in the Guangxi Province (Figure 1c). The rocks are of Yuxi Formation and comprise metamorphic argillaceous siltstone, metamorphic siltstone, slate, fine metamorphosed sandstone with sericitization and phyllitization, mafic–felsic volcanic ejecta, and basic–acidic volcanic ejecta. The host rocks contain extensive horizontal jointing.

The mafic rocks form pod-shaped bodies with an average area of approximately 1.5 km$^2$ (Figures 1c and 2a). These rocks have been modified by ductile and brittle deformation, metamorphism, and hydrothermal alteration (Figure 2b–g). The sampled gabbro intrusion appears somewhere between dark green and dark gray–green, and contains joints and fissures (Figure 2b,c). The gabbro samples record amphibolite facies metamorphism and display palimpsest textures (Figure 2e). The major minerals are plagioclase (40–50 vol.%), hornblende (30–40 vol.%), biotite (5–10 vol.%), chlorite (3–10 vol.%) and a small amount of biotite, calcite, quartz, with fine grain to blastodiabasic texture, massive structure, foliation structure (Figure 3a–c). The basic plagioclase is semi-idiomorphic column with obvious secondary changes, most of which are replaced by oblique zoisite, epidote and calcite, and biotite is distributed in fine scales (Figure 3a–c). The rocks are strongly fragmented and sheared in regions of mineralization. Microscopically, mica, chlorite, sericite, and quartz are significantly broken, and the altered minerals are oriented (Figure 3b,c). Post-hydrothermal quartz occurs in cracks in fragmented quartz grains, and pressure shadows were discovered in minerals with static recrystallization evidence (Figure 3c).

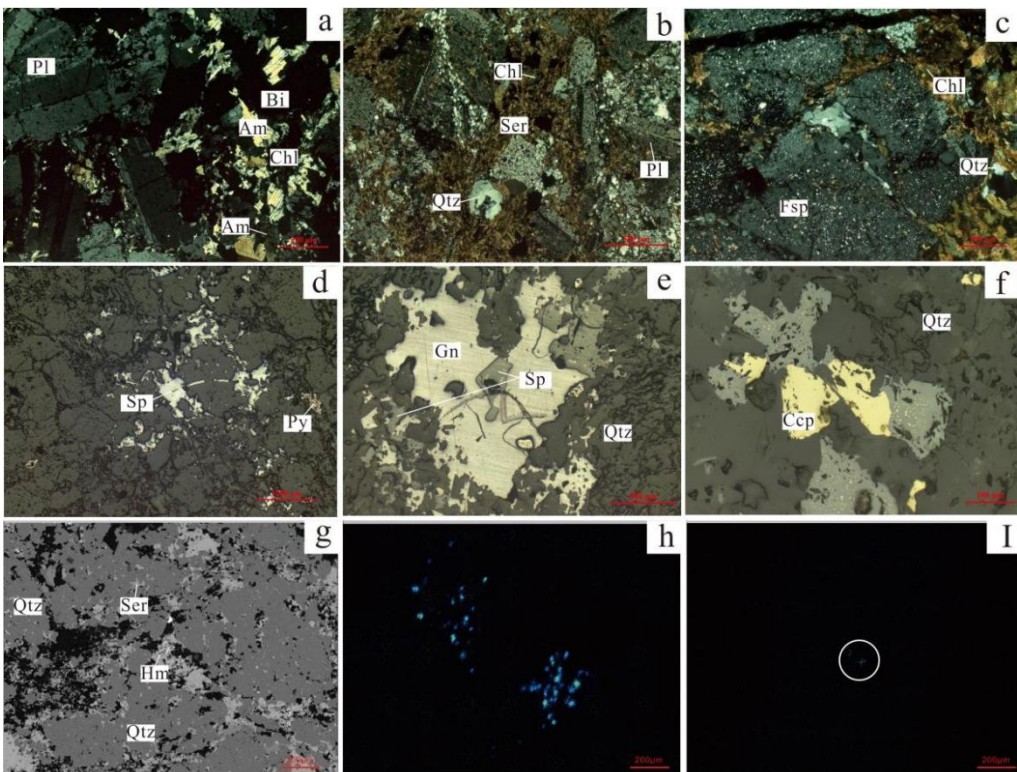

**Figure 3.** (**a–c**) Photomicrographs of fragmented and altered minerals (cross-polarized light); (**d–f**) photomicrographs of ore metasomatism (plane-polarized light); (**g**) morphology of minerals; and (**h,i**) cassiterite distribution map obtained using SEM.

The study area contains the NNE- and NNW-striking faults. The NNE-striking faults dip by 78° toward 290° and have damage zones 0.5–3.0 m wide. The NNW-striking faults developed later, and dip at an average angle of 82° toward 105° and have damage zones measuring 1–2 m in width. The fracturing of host rocks is common in the Ping'an area. The

dominant fractures dip by 62° toward 105° and 82° toward 261°. The subordinate fractures dip by 39° toward 278° and 81° toward 185°.

## 3. Sample Descriptions and Analytical Methodology

### 3.1. Sample Descriptions

Eight fresh gabbro rock samples and seven strongly hydrothermally altered gabbro samples obtained from ore bodies were collected from outcrops for geochemical analysis. Fresh rock samples obtained at 80–200 m from the ore body is gray–green with a large structure and generally intact feldspar quartz particles that are visible under a microscope. Within 10 m of the ore body, the altered gabbros have an apparent alteration halo and a substantial number of altered chlorite and other minerals that appear aligned under the microscope.

### 3.2. Whole-Rock Major- and Trace-Element Analyses

Whole-rock major- and trace-element analyses were performed using an X-ray fluorescence spectrometer and a Finnigan Element II inductively coupled plasma–mass spectrometer (ICP–MS) at the ICP–MS Laboratory of Guilin University of Technology, Guilin, China. The analytical precision is generally better than 2% for major elements and 10% for most trace elements. Sample solution preparation and analytical procedures were similar to those described by Li [38] and Zhang [39,40].

### 3.3. Zircon U–Pb Dating

Whole samples (>50 kg) were crushed, following which zircons were separated using standard heavy liquid and magnetic techniques, and then the samples were handpicked under a binocular microscope. Zircon grains of different color, size, and morphology were selected and mounted in epoxy in circular grain mounts and then ground to approximately half-grain thickness. The zircon U–Pb age determinations were performed at the Guangxi Key Laboratory of Hidden Metal Mineral Exploration, Guilin University of Technology, Guilin, China. The zircon U–Pb ages were determined using a laser-ablation (LA)–ICP–MS instrument (Agilent 7500cx), with a laser beam diameter of 32 μm, repetition rate of 6 Hz, and fluence of 2.5 J/cm$^2$. To ensure the reliability of analyses and stability of the instrument, standard materials were analyzed before and after analyzing unknown samples using the National Institute for Standards and Technology Standard Reference Material (NIST SRM) 610 and ARC National Key Center for Geochemical Evolution and Metallogeny of Continents standard zircon GJ-1. During the analyses, two determinations of each reference material were performed for every 10 sample spots tested. The measured data were processed using ICPMSDataCal109 software, and U–Pb Concordia diagrams were drawn using Isoplot v4.15 [41–43].

## 4. Results

### 4.1. Ore Body Characteristics

The whole Pb–Zn–Cu ore body occurs in a zonal pattern. Most of the ore body is disseminated at the bottom of altered gabbro; a small part is veined at the top of gabbro, and the ore body is mainly controlled by the joints of dip angles of 56°–70° and dips towards of 100°–120° or quartz veins, which are mainly distributed in gabbro (Figure 2a). According to field observations, several quartz veins approximately 10–15 m in length and 2–15 cm in width exist in ore bodies. The ore body mainly occurs in the strong alteration zone of gabbro, and there are obvious metasomatic phenomena along the rock joints and fissures (Figure 2b–e). The ore body strikes the NNE, not extending far and ranging in width from 4 to 10 cm. A field rapid analyzer test found that Pb, Zn, Cu, Sn, and other metallic elements reached or approached industrial grade in the area of strong alteration. The Ping'an deposit mainly has two types of mineralized structures: one is a quartz vein system with local faults, and other is the hydrothermal mineralization of joint fissure development, mainly for the hydrothermal mineralization with invasion of structural fissure filling, ore mineral

metasomatic and, according to the mineral analyzer test, visible only in the quartz veins is an obvious mineralization reaction. Close to a quartz vein, the ore-body Pb–Zn–Sn ore grade is significantly richer; namely, when quartz vein is near itself, the lead–zinc mineralization degree is high and gradually weakened from the vein body to the rock mass. The ore-controlling mechanism is mainly in the Sn direction along shear deformation, and Pb–Zn–Sn polymetallic–ore bodies are exposed along the joint direction of Minniang (Figure 2h). The mineralization body zoning is obvious, the ore body–extension length is short, the width of the ore body is large, and the distribution of the mineralized body is scattered. The main metallic elements in the ore are Pb, Zn, Cu, and Sn, as well as Fe, Ti, Co, and Ni. The grades of Pb, Zn, and Cu in the ore are 0.3–0.85%, 0.25–0.56%, and 0.08–0.15%, respectively. The main ore minerals are sphalerite, chalcopyrite, galena, and Cassiterite. The sphalerite develops in a small amount in the metal sulfide in disseminated form and has a high degree of self-shape, a large crystal grain size, a pale-yellow color, and good transparency. Further, minerals such as sphalerite crystallize along the flow direction (Figure 3d,e), and a small amount of milky pyrite is separated in the sphalerite crystal (Figure 3d). The chalcopyrite is the main metal mineral, with a bright brass color. Its crystal form is mainly semi-hydromorphic. It is symbiotic with pyrite in the mineral fissures in the shape of star points, and it has metallic luster under reflected light (Figure 3e,f). The galena is euhedral, other-shaped, agglomerated, and has a disseminated structure. A part of the euhedral galena is present in the sphalerite (Figure 3e). A tin deposit is located 2 km to the south of mining area. The tin content has reached 0.1% in rock and geochemical tests, but no cassiterite was found with the naked eye or when using a microscope. A scanning electron microscope (SEM) was used to detect ilmenite and surrounding samples, and Sn was found to exist in two states. One of them is an independent mineral cassiterite with a diameter of less than 2 μm (Figure 3h,i). Figure 3h shows that Sn is mainly distributed in the pores of ilmenite and the surrounding cracks in the form of stars (Figure 3h). The other is in ilmenite, which is cloudy and relatively uniform (Figure 3i), indicating the existence of homogeneous isomorphism. Figure 3i shows that cassiterite particles in ilmenite are extremely small. These characteristics indicate that tin mainly exists in mineral cracks in the form of extremely small cassiterite particles and a small amount exists in ilmenite in the form of homogeneous isomorphism.

### 4.2. Major and Trace Elements

#### 4.2.1. Fresh Gabbro Sample

The gabbros have $SiO_2$ contents between 46.85 and 52.23 wt.% (mean 49.89 wt.%), $K_2O$ between 0.72 and 1.53 wt.% (mean 0.95 wt.%), and $Na_2O$ between 0.79 and 3.93 wt.% (mean 2.42 wt.%). The $Na_2O + K_2O$ content is between 1.52 and 4.75 wt.%; thus, the samples can be classified as alkaline gabbro in a $(Na_2O + K_2O)$ vs. $SiO_2$ discrimination diagram (Figure 4). The gabbros are characterized by enrichment of rare earth elements (REEs; 57.21–80.56 ppm) and weaker enrichment of light REEs (LREEs) than heavy REEs (HREEs). The samples show (La/Yb)CN (where CN indicates normalized to chondrite) ratios between 3.03 and 3.63, (Gd/Yb)CN ratios between 0.98 and1.08, and slight negative Eu anomalies (Eu/Eu* = 0.75–0.93), similar to those of the average E-MORB (Figure 5). The trace-element variation diagrams (Figure 5) show that the fresh gabbros are characterized by moderate depletion in Nb and slight negative to positive Ti anomalies.

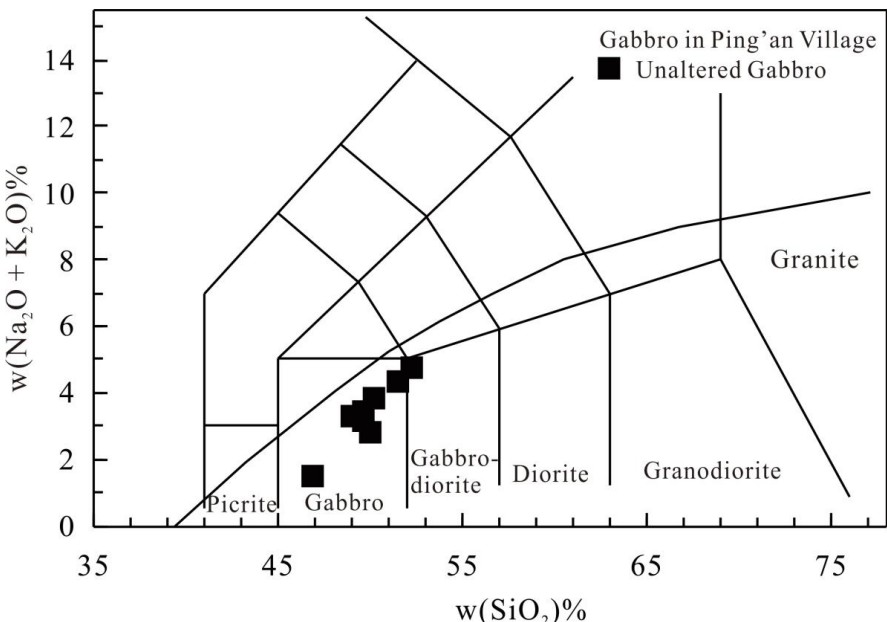

**Figure 4.** Total-alkalis–silica classification diagram for the fresh gabbros from the Ping'an Pb–Zn–Cu deposit.

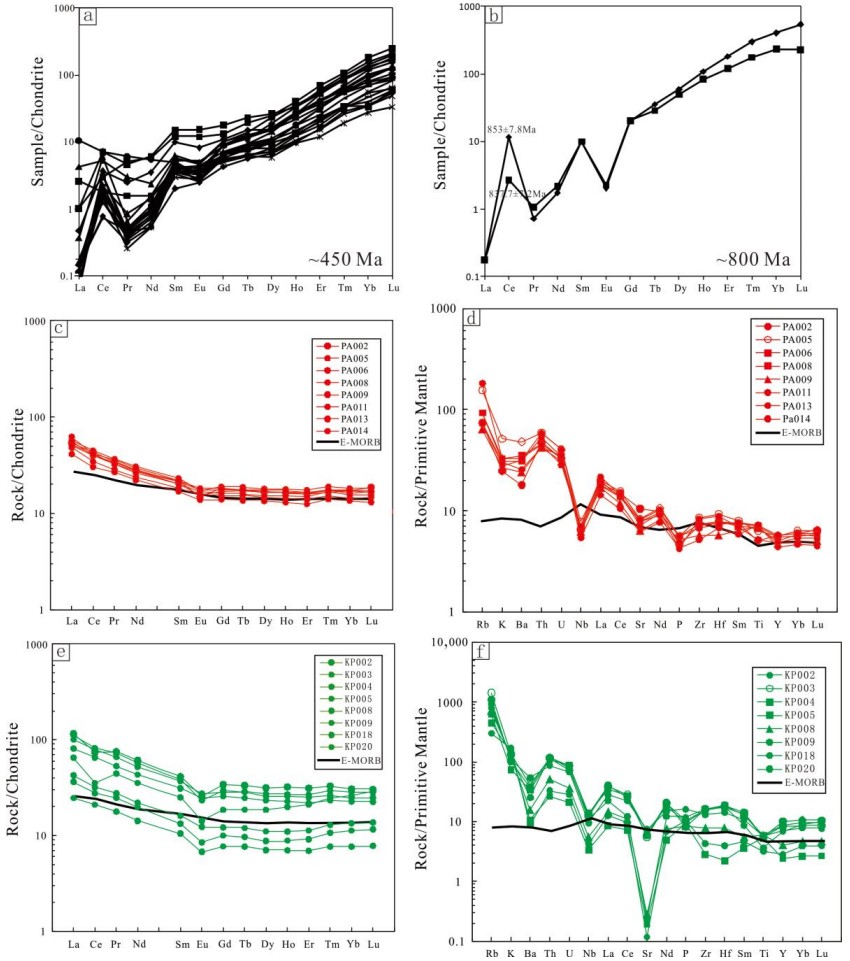

**Figure 5.** The corresponding chondrite–normalized distribution of the REE in the zircon crystals (**a**,**b**); (**c**,**d**) unaltered gabbro; and (**e**,**f**) altered gabbro.

#### 4.2.2. Altered Gabbro Samples

The altered gabbro contains higher $SiO_2$ (mean 56.63 wt.%) and $Al_2O_3$ (mean 15.86 wt.%) and lesser $Fe_2O_3$, MgO, and CaO contents than those in the gabbro (Table 1). The altered gabbro also contains higher Rb and lesser Sr contents than those in the gabbro. The Pb, Zn, and Sn contents in altered gabbro are also higher than in the gabbro (Table 1). These patterns reflect the enrichment of elements in the altered gabbros caused by the hydrothermal processes and alteration of minerals in the original rocks.

**Table 1.** Geochemical data for the studied samples.

| No. | Altered Gabbro | | | | | | | | Gabbro | | | | | | | |
|---|---|---|---|---|---|---|---|---|---|---|---|---|---|---|---|---|
| | KP002 | KP003 | KP004 | KP005 | KP008 | KP017 | KP018 | KP020 | PA002 | PA005 | PA006 | PA008 | PA009 | PA011 | PA013 | PA014 |
| $SiO_2$ | 58.03 | 54.41 | 53.57 | 54.47 | 56.11 | 57.45 | 59.56 | 59.44 | 46.85 | 51.54 | 50.31 | 49.58 | 49.64 | 52.23 | 48.99 | 50.01 |
| $TiO_2$ | 1.20 | 1.17 | 1.25 | 1.13 | 1.26 | 1.24 | 0.78 | 0.68 | 1.08 | 1.35 | 1.43 | 1.49 | 1.5 | 1.1 | 1.52 | 1.54 |
| $Al_2O_3$ | 15.88 | 15.41 | 16.12 | 15.63 | 16.45 | 15.35 | 15.52 | 16.49 | 11.75 | 13.28 | 13.29 | 13.2 | 13.7 | 15.84 | 13.3 | 13.63 |
| $Fe_2O_3$ | 10.62 | 12.63 | 11.87 | 14.48 | 12.53 | 12.06 | 12.88 | 9.61 | 20.5 | 14.83 | 15.63 | 16.18 | 15.87 | 12.35 | 15.89 | 15.9 |
| MgO | 2.76 | 3.41 | 3.04 | 2.74 | 2.17 | 2.58 | 1.28 | 0.96 | 4.11 | 4.86 | 5.13 | 5.29 | 4.92 | 3.86 | 5.32 | 5.19 |
| CaO | 2.63 | 2.25 | 4.59 | 0.82 | 0.86 | 2.59 | 0.38 | 1.06 | 11.94 | 8.37 | 8.93 | 9.21 | 9.47 | 7.51 | 9.19 | 9 |
| $K_2O$ | 3.07 | 3.95 | 2.19 | 4.12 | 3.55 | 3.23 | 3.53 | 4.99 | 0.73 | 1.53 | 0.92 | 0.97 | 0.79 | 0.82 | 0.88 | 0.98 |
| MnO | 0.24 | 0.26 | 0.22 | 0.46 | 0.35 | 0.21 | 0.17 | 0.25 | 0.24 | 0.2 | 0.21 | 0.21 | 0.22 | 0.17 | 0.21 | 0.24 |
| $Na_2O$ | 5.04 | 4.04 | 5.30 | 4.34 | 5.31 | 3.79 | 4.10 | 4.88 | 0.79 | 2.79 | 2.75 | 2.44 | 2.4 | 3.93 | 2.39 | 1.86 |
| $P_2O_5$ | 0.18 | 0.22 | 0.20 | 0.20 | 0.18 | 0.35 | 0.25 | 0.22 | 0.1 | 0.12 | 0.12 | 0.1 | 0.11 | 0.09 | 0.09 | 0.12 |
| LOI | 1.00 | 1.39 | 1.21 | 1.28 | 1.15 | 1.20 | 1.41 | 1.26 | 0.97 | 0.95 | 1.16 | 1.33 | 1.23 | 1.39 | 1.43 | 1.68 |
| Total | 100.65 | 99.13 | 99.55 | 99.67 | 99.92 | 100.05 | 99.88 | 99.82 | 99.07 | 99.82 | 99.87 | 100.01 | 99.85 | 99.29 | 99.21 | 100.14 |
| Sn | 136.56 | 114.01 | 184.02 | 39.38 | 65.77 | 88.05 | 76.42 | 19.87 | 6.64 | 2.4 | 2.15 | 3.76 | 12.08 | 15.85 | 9.39 | 7.78 |
| Pb | 1036.47 | 1722.38 | 916.46 | 5286.83 | 3590.28 | 816.54 | 2324 | 5504.96 | 15.18 | 8.59 | 7.94 | 9.52 | 14.88 | 13.07 | 10.18 | 13.5 |
| Zn | 1465.07 | 845.45 | 1395.6 | 3219.15 | 1776.37 | 237.03 | 341.25 | 2698.03 | 149.37 | 149.73 | 138.27 | 182.29 | 178.6 | 164.5 | 135.65 | 199.93 |
| Mo | 0 | 0 | 0.01 | 0 | 0 | 0.08 | 0.06 | 0 | 0.26 | 0.32 | 0.22 | 0.37 | 0.21 | 0.08 | 0.3 | 0.2 |
| Cu | 7.55 | 2.87 | 2.2 | 7.34 | 10.28 | 33.12 | 34.13 | 113.61 | 711.27 | 203.98 | 292.85 | 294.88 | 594.11 | 123.61 | 299.9 | 422.65 |
| W | 22.63 | 20.86 | 21.77 | 66.47 | 54.92 | 22.11 | 34.67 | 37.89 | 26.16 | 75.62 | 30.44 | 29.4 | 26.73 | 31.74 | 25.01 | 33.71 |
| La | 27.34 | 26.46 | 23.88 | 5.85 | 10.11 | 19.23 | 15.43 | 8.64 | 12.80 | 13.58 | 12.23 | 13.41 | 11.46 | 11.98 | 9.79 | 14.80 |
| Ce | 45.52 | 49.87 | 48.62 | 12.88 | 19.51 | 39.99 | 21.62 | 16.91 | 26.47 | 27.55 | 25.32 | 24.21 | 21.12 | 24.47 | 18.55 | 25.27 |
| Pr | 7.14 | 6.88 | 6.26 | 1.69 | 2.62 | 5.04 | 4.21 | 2.36 | 3.27 | 3.49 | 3.10 | 3.25 | 2.72 | 3.09 | 2.54 | 3.44 |
| Nd | 28.37 | 26.58 | 24.39 | 6.65 | 10.24 | 20.25 | 16.56 | 9.26 | 13.26 | 14.31 | 12.73 | 12.87 | 11.12 | 12.30 | 10.36 | 13.85 |
| Sm | 6.30 | 5.87 | 5.66 | 1.61 | 2.47 | 4.75 | 3.83 | 2.04 | 3.31 | 3.52 | 3.11 | 3.29 | 2.82 | 3.08 | 2.58 | 3.37 |
| Eu | 1.49 | 1.35 | 1.59 | 0.39 | 0.71 | 1.44 | 0.82 | 0.49 | 0.86 | 1.00 | 0.91 | 1.02 | 0.90 | 0.98 | 0.81 | 1.06 |
| Gd | 7.02 | 6.10 | 5.81 | 1.59 | 2.50 | 5.30 | 3.82 | 2.05 | 3.74 | 3.87 | 3.43 | 3.62 | 3.13 | 3.32 | 2.87 | 3.70 |
| Tb | 1.25 | 1.06 | 1.08 | 0.29 | 0.45 | 0.92 | 0.70 | 0.36 | 0.65 | 0.70 | 0.65 | 0.66 | 0.58 | 0.60 | 0.51 | 0.66 |
| Dy | 7.98 | 6.54 | 6.94 | 1.79 | 2.81 | 5.92 | 4.72 | 2.23 | 4.28 | 4.57 | 4.17 | 4.31 | 3.72 | 3.88 | 3.41 | 4.44 |
| Ho | 1.82 | 1.45 | 1.52 | 0.40 | 0.62 | 1.29 | 1.13 | 0.50 | 0.93 | 1.01 | 0.92 | 0.94 | 0.81 | 0.87 | 0.73 | 0.97 |
| Er | 5.17 | 4.15 | 4.43 | 1.15 | 1.86 | 3.62 | 3.51 | 1.51 | 2.67 | 2.89 | 2.60 | 2.62 | 2.32 | 2.49 | 2.07 | 2.76 |
| Tm | 0.84 | 0.67 | 0.75 | 0.20 | 0.33 | 0.60 | 0.63 | 0.27 | 0.44 | 0.48 | 0.42 | 0.45 | 0.39 | 0.43 | 0.36 | 0.45 |
| Yb | 5.20 | 4.21 | 4.81 | 1.30 | 2.29 | 3.85 | 4.44 | 1.92 | 2.93 | 3.11 | 2.90 | 2.77 | 2.45 | 2.67 | 2.30 | 2.92 |
| Lu | 0.77 | 0.63 | 0.72 | 0.20 | 0.35 | 0.57 | 0.74 | 0.29 | 0.47 | 0.47 | 0.47 | 0.42 | 0.43 | 0.38 | 0.39 | 0.43 |
| Ba | 259.90 | 285.68 | 237.20 | 73.63 | 107.71 | 376.21 | 175.53 | 60.78 | 125.57 | 336.62 | 215.49 | 246.21 | 166.34 | 218.70 | 178.84 | 231.62 |
| Th | 9.84 | 9.73 | 9.38 | 2.23 | 4.31 | 7.37 | 10.21 | 2.80 | 4.04 | 4.99 | 4.54 | 3.85 | 3.55 | 4.97 | 3.57 | 3.79 |
| Ta | 0.76 | 0.70 | 0.79 | 0.18 | 0.32 | 0.57 | 0.79 | 0.23 | 0.40 | 0.79 | 0.39 | 0.46 | 0.35 | 0.40 | 0.30 | 0.39 |
| Cr | 0.77 | 1.70 | 1.79 | 0.34 | 0.88 | 0.92 | 0.51 | 0.00 | 12.21 | 10.42 | 11.83 | 11.21 | 7.45 | 1.46 | 8.00 | 9.14 |
| Ni | 5.24 | 6.44 | 5.27 | 1.33 | 1.65 | 3.11 | 0.77 | 0.04 | 39.36 | 40.75 | 40.23 | 42.78 | 33.79 | 21.33 | 37.24 | 37.25 |
| Rb | 688.05 | 896.95 | 393.12 | 285.06 | 445.85 | 582.02 | 500.40 | 188.74 | 46.88 | 98.47 | 58.98 | 58.97 | 40.36 | 47.68 | 46.52 | 115.35 |
| Sr | 123.71 | 116.32 | 144.61 | 4.07 | 5.91 | 156.40 | 4.84 | 2.50 | 220.82 | 173.43 | 150.46 | 163.36 | 133.34 | 164.83 | 130.96 | 172.19 |
| Hf | 5.73 | 5.32 | 5.68 | 0.69 | 2.40 | 4.42 | 5.55 | 1.22 | 2.41 | 2.86 | 2.42 | 2.29 | 1.77 | 2.71 | 2.09 | 2.27 |
| U | 1.82 | 1.50 | 1.82 | 0.44 | 0.76 | 1.41 | 1.61 | 0.59 | 0.69 | 0.84 | 0.72 | 0.69 | 0.64 | 0.84 | 0.74 | 0.59 |
| Nb | 9.50 | 8.80 | 9.80 | 2.40 | 3.94 | 6.76 | 7.97 | 3.04 | 4.62 | 5.53 | 4.73 | 4.97 | 4.42 | 4.82 | 3.84 | 5.03 |
| Zr | 181.56 | 174.72 | 177.59 | 31.72 | 86.46 | 145.82 | 178.80 | 48.14 | 80.78 | 95.47 | 81.74 | 79.87 | 63.56 | 92.21 | 57.52 | 75.18 |
| Co | 28.98 | 33.39 | 37.38 | 9.63 | 13.20 | 29.59 | 6.65 | 5.87 | 61.02 | 51.69 | 56.45 | 55.54 | 49.09 | 46.60 | 51.71 | 52.53 |
| Sc | 30.29 | 31.83 | 24.12 | 11.64 | 19.93 | 26.80 | 9.50 | 4.48 | 45.67 | 51.80 | 15.60 | 65.07 | 56.90 | 1.93 | 51.86 | 63.70 |

### 4.3. Zircon U–Pb Dating from Altered Gabbro

The zircons separated from 16 ore samples were dated using U–Pb geochronology. Zircon ages with high concordance (>90%) were retained and low concordance were disregarded in age calculations. The 16 zircons were then divided into two groups according to age: Group I and Group II (Table 2).

**Table 2.** Results of LA–ICP–MS zircon U–Pb age determinations.

| | Pb (ppm) | Th (ppm) | U (ppm) | Th/U | Ratios | | | | Age | | | | Concordance |
|---|---|---|---|---|---|---|---|---|---|---|---|---|---|
| | | | | | $^{207}Pb/^{235}U$ | 1σ | $^{206}Pb/^{238}U$ | 1σ | $^{207}Pb/^{235}U$ | 1σ | $^{206}Pb/^{238}U$ | 1σ | |
| PA1-01 | 293 | 978 | 1115 | 0.88 | 0.602159 | 0.019082 | 0.073987 | 0.000861 | 478.6 | 12.1 | 460.1 | 5.2 | 96% |
| PA1-03 | 56 | 140 | 280 | 0.50 | 0.618039 | 0.027514 | 0.074253 | 0.000982 | 488.6 | 17.3 | 461.7 | 5.9 | 94% |
| PA1-04 | 329 | 1262 | 1071 | 1.18 | 0.573571 | 0.016010 | 0.075000 | 0.000780 | 460.3 | 10.3 | 466.2 | 4.7 | 98% |
| PA1-06 | 178 | 559 | 944 | 0.59 | 0.539200 | 0.013967 | 0.073515 | 0.000835 | 437.9 | 9.2 | 457.3 | 5.0 | 95% |
| PA1-07 | 66 | 202 | 379 | 0.53 | 0.593061 | 0.024638 | 0.070920 | 0.000962 | 472.8 | 15.7 | 441.7 | 5.8 | 93% |
| PA1-08 | 146 | 438 | 815 | 0.54 | 0.572501 | 0.016237 | 0.072294 | 0.000725 | 459.6 | 10.5 | 450.0 | 4.4 | 97% |
| PA1-11 | 106 | 331 | 559 | 0.59 | 0.580340 | 0.022677 | 0.073969 | 0.000920 | 464.7 | 14.6 | 460.0 | 5.5 | 98% |
| PA1-12 | 89 | 196 | 568 | 0.34 | 0.652880 | 0.021106 | 0.078289 | 0.000873 | 510.2 | 13.0 | 485.9 | 5.2 | 95% |
| PA1-14 | 284 | 1092 | 1086 | 1.01 | 0.612011 | 0.017282 | 0.071839 | 0.000702 | 484.8 | 10.9 | 447.2 | 4.2 | 91% |
| PA1-15 | 64 | 216 | 335 | 0.64 | 0.580762 | 0.023402 | 0.072154 | 0.000968 | 465.0 | 15.0 | 449.1 | 5.8 | 96% |
| PA1-17 | 93 | 318 | 489 | 0.65 | 0.551845 | 0.020783 | 0.069325 | 0.000810 | 446.2 | 13.6 | 432.1 | 4.9 | 96% |
| PA1-19 | 56 | 172 | 311 | 0.55 | 0.566945 | 0.026136 | 0.072613 | 0.001032 | 456.0 | 16.9 | 451.9 | 6.2 | 99% |
| PA1-20 | 200 | 814 | 885 | 0.92 | 0.562974 | 0.017154 | 0.069289 | 0.000737 | 453.5 | 11.1 | 431.9 | 4.4 | 95% |
| PA1-21 | 204 | 682 | 949 | 0.72 | 0.609279 | 0.017887 | 0.073101 | 0.000827 | 483.1 | 11.3 | 454.8 | 5.0 | 93% |
| PA1-24 | 1450 | 3538 | 1526 | 2.32 | 1.338613 | 0.028496 | 0.141470 | 0.001385 | 862.6 | 12.4 | 853.0 | 7.8 | 98% |
| PA1-25 | 2351 | 5656 | 2077 | 2.72 | 1.310155 | 0.026897 | 0.138766 | 0.001265 | 850.2 | 11.8 | 837.7 | 7.2 | 98% |

Group I zircons (two samples) have lengths between 80 and 110 μm, widths between 78 and 100 μm, and aspect ratios of 1:1. They are round in shape and show bright rims under cathodoluminescence imaging. The cores exhibit oscillatory zoning, a dark gray color under cathodoluminescence imaging, and Th/U ratios of 2.32 and 2.72. The analyses yielded $^{206}Pb/^{238}U$ ages of 850 ± 7.8–837.7 ± 7.2 Ma, similar to the age of the Neoproterozoic magmatic intrusions.

Group II zircons are mainly short prismatic, 70–200 μm long, and 65–80 μm wide, with aspect ratios of 2:1. These zircons are euhedral to subhedral and are transparent and colorless. The zircon rims show oscillatory zoning and smooth surfaces and contain few fluid inclusions. All these characteristics are consistent with those of magmatic zircons and differ from those of hydrothermal or metamorphic zircons [44]. The Th contents of Group II zircon range from 140 to 1262 ppm and of U from 311 to 2358 ppm. The Th/U ratio varies from 0.34 to 1.18. Zircon U-Pb data for the Jiufeng tin deposit cassiterite is presented using a Tera–Wasserburg diagram (Figure 6). $^{206}Pb/^{238}U$ ages range from 466.2 to 431.9 Ma, with a weighted mean age of 450.4 ± 6.7 Ma (1σ, MSWD = 4.9 and *n* = 14).

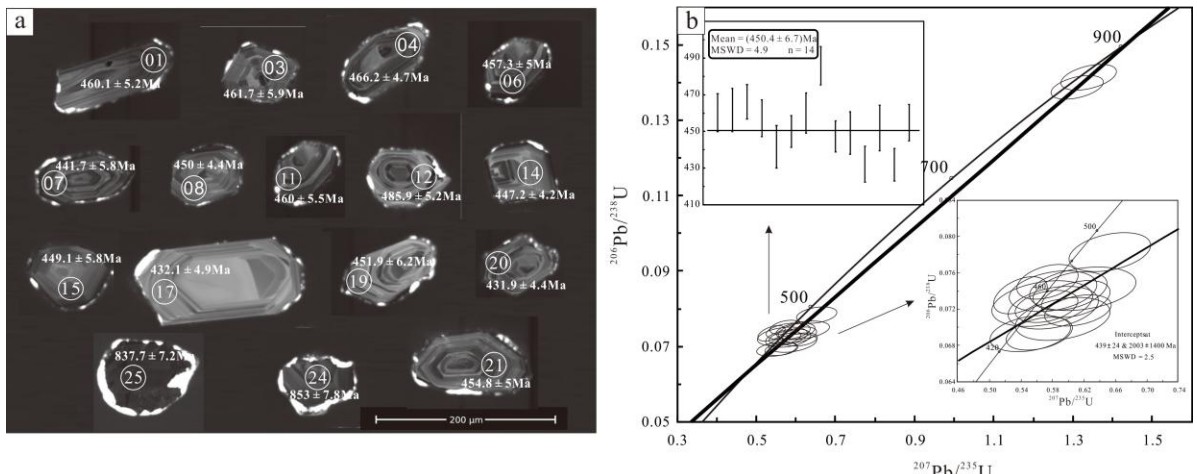

**Figure 6.** (**a**) Representative cathodoluminescence images of zircons obtained from cassiterite (ore specimen PA01) from the Ping'an Pb–Zn–Cu deposit. White circles with numbers indicate spots used for LA–ICP–MS U–Pb analysis. (**b**) Tera–Wasserburg diagram of LA–ICP–MS U–Pb data for zircons obtained from cassiterite (ore specimen PA01) from the Ping'an Pb–Zn–Cu deposit.

### 4.4. Zircon Trace-Element Contents

The trace elements in zircon can be used to identify its origin. In general, the Hf contents range from 5.1 to 7.6 × 10$^{-6}$ and the Nb/Ta ratio is >1 in magmatic zircons [45,46]. In this study (Table 3), the Hf contents of zircons vary from 2.14 to 2.68 and the Nb/Ta ratio

varies between 7 and 13.33, with a mean of 11.88, which is in the range of magmatic zircon. The pronounced positive Ce anomalies, relatively low total REE contents ($803–8893 \times 10^{-6}$), and REE patterns are different from those of enriched REEs that are typical of hydrothermal zircon. Further, they have no obvious negative anomalies in Ce or Eu. The magmatic zircons from mantle regions commonly have low total REE contents and weak negative or no Eu anomalies, whereas the magmatic zircons from crustal regions have low trace-element contents and strong positive Ce and negative Eu anomalies [47,48]. Group I zircons have Y contents of 12,263–12,526 ppm and Yb contents of 3787–3788 ppm, as well as marked positive Ce anomalies and negative Eu anomalies. Group II zircons have total REE contents of 8266–8893 ppm with a mean of $1405 \times 10^{-6}$, Y contents of 886–2547 with a mean of $1655 \times 10^{-6}$, Yb contents of $326–868 \times 10^{-6}$ with a mean of $643 \times 10^{-6}$, and weak negative Eu anomalies [49–51]. These data suggest that Group I zircons were derived from felsic magma and Group II zircons from mafic magma.

**Table 3.** Geochemical data for the studied zircons.

|      | La   | Ce   | Pr   | Nd   | Sm   | Eu   | Gd   | Tb   | Dy    | Ho   | Er    | Tm   | Yb    | Lu    | Y      |
|------|------|------|------|------|------|------|------|------|-------|------|-------|------|-------|-------|--------|
| PA01 | 0.11 | 2.27 | 0.24 | 1.65 | 1.53 | 0.48 | 2.27 | 0.56 | 3.93  | 1.41 | 6.54  | 1.75 | 20.18 | 3.93  | 36.97  |
| PA03 | 0.62 | 1.09 | 0.15 | 0.72 | 0.64 | 0.20 | 1.34 | 0.30 | 2.07  | 0.78 | 3.79  | 0.86 | 5.74  | 1.51  | 13.65  |
| PA04 | 0.09 | 3.67 | 0.06 | 0.73 | 0.88 | 0.27 | 2.07 | 0.52 | 5.91  | 1.99 | 9.35  | 2.28 | 22.54 | 4.36  | 63.45  |
| PA06 | 0.01 | 1.92 | 0.04 | 0.34 | 0.46 | 0.16 | 1.37 | 0.35 | 3.58  | 1.49 | 8.44  | 2.23 | 23.84 | 5.02  | 48.05  |
| PA07 | 2.49 | 4.42 | 0.58 | 2.55 | 0.76 | 0.20 | 1.20 | 0.29 | 2.53  | 1.04 | 5.24  | 1.55 | 16.42 | 3.16  | 29.59  |
| PA08 | 0.04 | 0.78 | 0.04 | 0.37 | 0.51 | 0.15 | 1.36 | 0.34 | 2.72  | 0.84 | 3.59  | 0.87 | 9.37  | 2.11  | 19.94  |
| PA11 | 0.02 | 1.40 | 0.04 | 0.39 | 0.54 | 0.19 | 1.09 | 0.25 | 1.84  | 0.70 | 2.95  | 0.84 | 8.00  | 1.67  | 16.26  |
| PA12 | 0.04 | 0.48 | 0.03 | 0.25 | 0.31 | 0.14 | 0.88 | 0.21 | 1.71  | 0.56 | 2.61  | 0.74 | 6.09  | 1.38  | 11.66  |
| PA14 | 1.02 | 3.27 | 0.28 | 1.11 | 0.97 | 0.28 | 2.09 | 0.47 | 3.71  | 1.48 | 6.95  | 1.59 | 17.34 | 3.35  | 41.21  |
| PA15 | 0.00 | 0.79 | 0.04 | 0.35 | 0.63 | 0.16 | 1.15 | 0.23 | 1.83  | 0.59 | 2.55  | 0.68 | 5.88  | 1.24  | 13.52  |
| PA17 | 0.02 | 1.01 | 0.05 | 0.52 | 0.79 | 0.18 | 1.81 | 0.46 | 4.10  | 1.56 | 6.72  | 1.55 | 13.20 | 2.35  | 38.28  |
| PA19 | 0.00 | 1.15 | 0.04 | 0.25 | 0.55 | 0.21 | 1.33 | 0.28 | 2.44  | 0.71 | 3.38  | 0.88 | 6.88  | 1.51  | 16.46  |
| PA20 | 0.03 | 1.96 | 0.05 | 0.47 | 0.82 | 0.30 | 1.77 | 0.44 | 3.83  | 1.24 | 5.52  | 1.39 | 13.59 | 2.68  | 34.09  |
| PA21 | 0.03 | 1.99 | 0.05 | 0.41 | 0.55 | 0.23 | 1.84 | 0.42 | 4.74  | 1.91 | 9.94  | 2.52 | 26.47 | 5.30  | 59.47  |
| PA24 | 0.04 | 7.12 | 0.07 | 0.82 | 1.52 | 0.12 | 4.12 | 1.33 | 14.91 | 6.16 | 29.97 | 7.66 | 69.03 | 13.78 | 154.47 |
| PA25 | 0.04 | 1.65 | 0.10 | 1.02 | 1.52 | 0.13 | 4.17 | 1.07 | 12.57 | 4.72 | 19.84 | 4.47 | 39.68 | 5.80  | 116.36 |

## 5. Discussion

### 5.1. Magmatic Activity during the Caledonian Orogeny

#### 5.1.1. Timing of Magmatism

The Caledonian orogeny is referred to as the Guangxi orogeny in Guangxi Province [52]. The Caledonian magmatic rocks are common in the eastern, southern, and central Guangxi Province [2,34,53]. However, Caledonian magmatic activity has rarely been reported in the northern Guangxi Province. Usually, it is considered that that during the Sibao Movement, the Yangtze and Cathaysian plates collided and spliced to form the Cathaysia continent, and then disintegrated with the breaking of the global Rodinia supercontinent until the Caledonian orogeny occurred to form the current South China continent. Combined with the above dating results and regional tectonic evolution history, the study area has been affected by the Sibao, Early Caledonian, and late Caledonian tectonic activities, and produced hydrothermal zircons of different ages.

The zircon age test mainly obtained two main age groups. One is 837–853 Ma and the other is 421.9–485.9 Ma, with an average weighted age of 450.4 ± 6.7 Ma. In contrast, the 850 Ma ± zircon has a complex internal structure. The zircon is elliptic, its cathodoluminescence image is dark, and its edge has a bright hydrothermal accretion resulting from the hydrothermal fluid transformation of post-diagenetic structure (Figure 7a,b). This suggests that the zircon has experienced the influence of tectonic thermal events after its formation. Therefore, the hydrothermal crystallization of metallic elements, such as Cu, Pb, and Zn, is probably brought about by shear hydrothermal activity after gabbro crystallization. Most of the tin deposits in northern Guangxi were formed simultaneously

during the diagenetic age of the Motianling and Yuanbaoshan rocks. Therefore, we believe that the ages of these zircons were captured from other rocks during the magmatic rise of gabbro formation. The 450 Ma-aged zircon is mainly characterized by magmatic zircons in oscillating belts and its edge has sporadic hydrothermal accretion resulting, which can be interpreted as the crystallization age (Caledonian or Late Ordovician) (Figure 7c,d). This suggests that the study area has experienced multiple periods of tectonic magmatic events. Therefore, we think that the Caledonian magmatic activity in this area formed a large number of mafic and ultrabasic rocks. In recent years, Qin and Zhu have obtained zircons with Caledonian magmatic characteristics (421.9 ± 7.8 and 420.2–422.6 Ma), which also show this phenomenon. At the same time, the precise deformation and metamorphism age of the Motianling ductile shear zone is 393–425 Ma [54,55], which is slightly later than our test age (450.4 ± 6.7 Ma). The hydrothermal zircon age obtained by Li and Qin is 441Ma ±, which indicates that the Caledonian hydrothermal deformation is closely related to shear deformation [10,11].

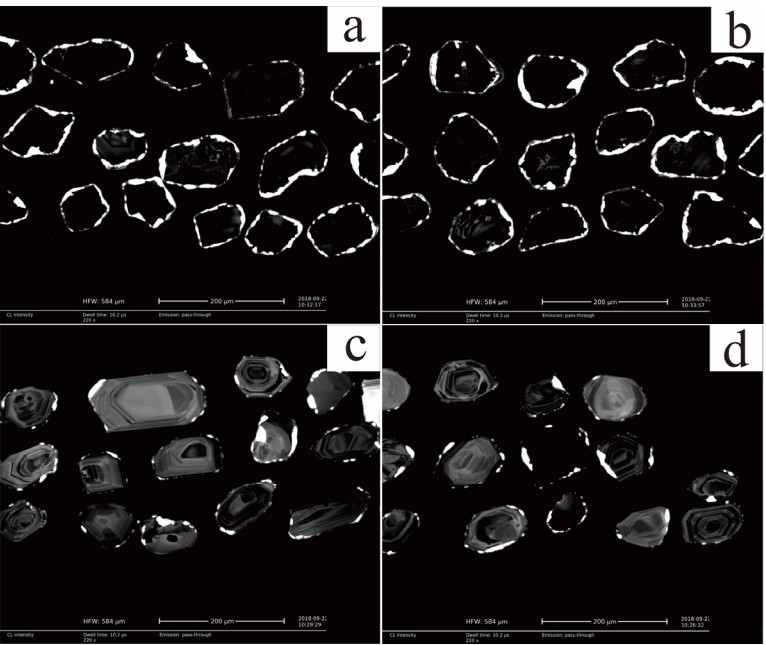

**Figure 7.** Cathodoluminescence images of the altered gabbro zircon from the gabbro: (**a**,**b**) Neoproterozoic zircon and (**c**,**d**) Caledonian zircon.

According to the authors, the late Caledonian age represents the age of brittle shear, the early Caledonian age corresponds to the age of hydrothermal invasion and filling, and the hydrothermal crystallization age containing Pb, Zn, and Cu elements indicates Caledonian-period-endogenous mineralization. The regional mineralization and hydrothermal activity are of high significance, so the authors believed that the hydrothermal zircon-aged mine is formed in the shear deformation age. At this time, the shear deformation in the region has ended and the deep hydrothermal system has not only obtained a large number of metal elements from three barrier rock masses, but also from the stratum (mainly Pb and Zn). The hydrothermal activity carried these metals onto the surface along the Sn-trending faults, forming the deposit.

### 5.1.2. Tectonic Setting of Ping'an Gabbros

The integration of field and petrographic observations and immobile-trace-element characteristics can be used to constrain the tectonic setting of gabbro formation. Generally, high-field-strength elements (HFSEs; e.g., Th, Nb, Ta, Zr, Hf, and Ti), REEs, and transition-metal elements (e.g., Sc, V, Cr, and Ni) are regarded as being relatively immobile during metamorphism and alteration. In contrast, some major elements (K, Na, and Ca) and large-ion lithophile elements (LILEs; e.g., Rb, Th, and U) are mobile. The similarity of

immobile elements between the studied gabbros and ores means that they can be used to analyze the tectonic setting of magmatism.

The gabbro is typically contaminated by crustal materials, as revealed by Proterozoic zircons captured by the ascending magma, weak differentiation of LREEs and HREEs, enrichment in some LILEs (Rb, Th, and U), and depletion in Nb and Ta. The degree of contamination in the mafic rocks is usually inferred from covariant relationships between Nb/La, Nb/Ce, and Nb/Ta [35,56]. For Ping'an gabbros, the lack of a relationship between Nb/Ta and La/Yb (not shown) indicates limited contamination. The ratios of Zr/Hf (mean of 11), Nb/U (mean of 12), and Nb/La (mean of 0.69) are lower than the average mantle values (36.27, 47, and 1, respectively) and average continental crust values (11, 12, and 0.7, respectively) [35,57–62]. Thus, the measured values cannot be explained by crustal contamination alone. The Nb/Zr versus Th/Zr (Figure 8) and Ba/Th versus Th/Nb (Figure 9) diagrams can be used to detect the signatures of subduction and metasomatism. The Nb/Ta ratio varies from 7 to 13.33 and has a mean of 11.8, which is lower than the chondrite value (17.6), suggesting the existence of subducted oceanic materials in the source area of magma that formed the Ping'an gabbros.

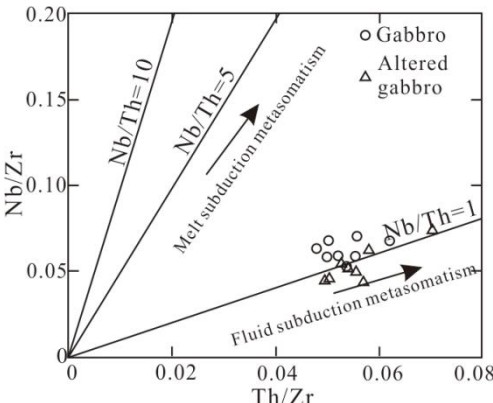

**Figure 8.** Th/Zr versus Nb/Zr discrimination diagram for Ping'an gabbros.

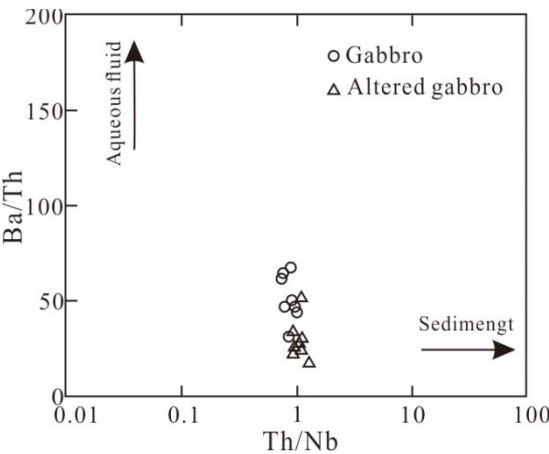

**Figure 9.** Ba/Th versus Th/Nb discrimination diagram for Ping'an gabbros.

In Th/Yb versus Ta/Yb (Figure 10) and Gb/Yb versus Sm (Figure 11) diagrams, all samples lie in the active continental margin field. In a Zr/Y versus Ti/Y diagram (Figure 12), all samples lie in the field of plate-margin basalts. These characteristics suggest a subduction-related setting similar to the setting interpreted for Neoproterozoic mafic–ultramafic rocks in the study region [58,62–64]. However, compared with these Neoproterozoic mafic–ultramafic rocks, the studied gabbros have higher FeO and lower MgO contents and no pronounced negative Eu and Ti anomalies.

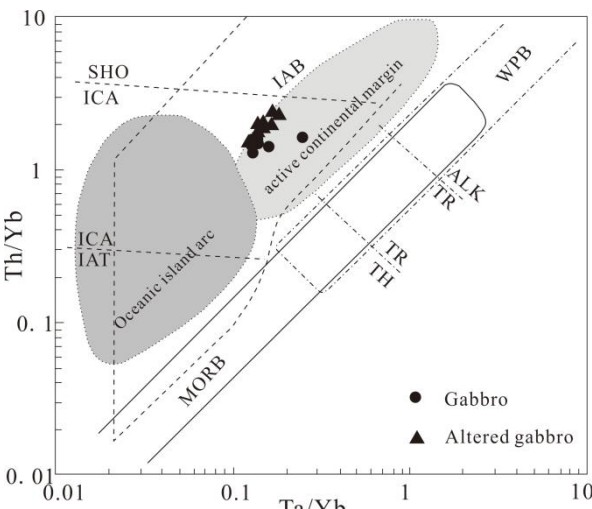

**Figure 10.** Th/Yb versus Ta/Yb discrimination diagram for Ping'an gabbros.

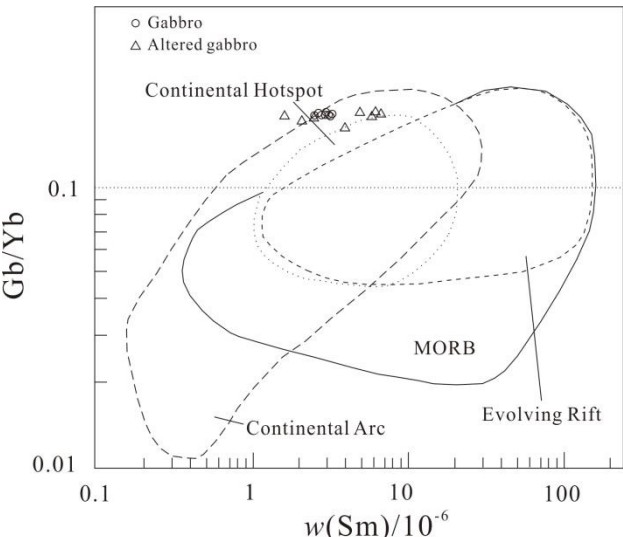

**Figure 11.** Gb/Yb versus Sm discrimination diagram for Ping'an gabbros.

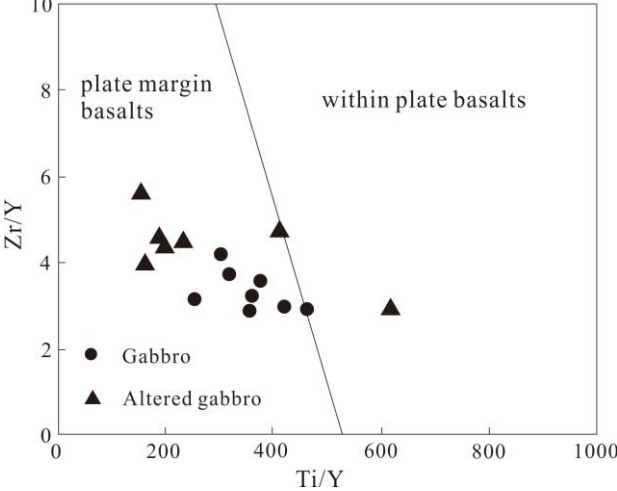

**Figure 12.** Zr/Y versus Ti/Y discrimination diagram for Ping'an gabbros.

Convergence and collision between the Yangtze and Cathaysian blocks or microplates began around 860–800 Ma [4,34,35,52]. Some scholars have proposed that the South China Block may not have amalgamated with the Cathaysian Block until the early Paleozoic age [11]. The research on gabbro in this paper supports the second view. The collision of nan-Guangxi-Beiyue, Yunkai, and Yangtze Blocks led to the Caledonian uplift of JOB, which resulted in low-grade metamorphism and intense granitic magmatism and mineralization in early gabbros. These are the product of the subduction of microplates during the Caledonian orogenesis [2,11].

### 5.2. Shear Structures in the Study Region

The Caledonian tectonic processes in the study region can be divided into N-directed compression, which resulted in early stage W-striking folds and faults, and NW-directed compression, which resulted in late-stage NNE-striking folds, shear zones, and faults (Figure 1b) [33,53]. The shear deformation can be divided into early stage NW-trending thrust compression (453–426 Ma) and late-stage NE-trending ductile shear (after 426 Ma) [16,65]. The Ar–Ar dating of mica minerals from the shear zones in the study region has yielded ages mainly in the range 425–393 Ma [15,36,54,66]. The dating of hydrothermal zircons from quartz veins has shown that the timing of their filling and alteration by hydrothermal activity was 445–420 Ma [11]. The age of gabbro (450 ± 6.7 Ma) determined in the present study marks the transition period between the two tectonic stress regimes, and this transition would have favored the ascent of mafic magma from deep regions. The age of studied gabbro is similar to that of the Maping kimberlite in Guizhou Province (U–Pb age of 438 Ma), which represents the South China Craton event [67].

The shear deformation during the Caledonian orogenesis led to the widespread development of shear joints in the study region. The two main groups of regional shear planes are NNE-striking and NNW-striking, with small and large dip angles, respectively. In the Ping'an area, joint-oriented dips toward 105° at 62° and toward 261° at 82° are part of the regional shear joint system.

After the intrusion of gabbros, regional shearing caused deformation through different modes, including fracturing at various scales, the development of preferred orientations of minerals (Figure 3a,c), and the formation of pressure shadows and bookshelf structures. The fractures, joints, and faults caused by shearing provided the space for fluid migration (Figure 3a–c). The shearing that occurred under the temperatures of 450–600 °C [12] led to the formation of hydrothermal fluids. The development of numerous quartz veins contemporaneously with rock alteration occurred when hydrothermal fluids filled the structural space and altered host rocks [4,16,52]. In summary, the late orogeneses in study region were characterized by small-scale tectonism and hydrothermal alteration, with shear controlling the formation and migration of hydrothermal fluids.

### 5.3. Metasomatic and Hydrothermal Fluids

In the Ping'an deposit, ore bodies in rock fissures were formed through the precipitation of minerals from hydrothermal fluids. Original minerals in gabbro show fracturing, whereas mafic minerals and feldspars have been altered and metasomatized to form biotite, chlorite, and epidote with preferential orientations (Figure 3a–c). Chalcopyrite, sphalerite, galena, and other metallic minerals are associated with quartz formed through hydrothermal crystallization. All these features suggest that the metal elements were formed by the migration of hydrothermal fluids into the surrounding gabbro through cracks formed by previous deformation. Macroscopically, ore bodies are filled in rock fissures and fracture structures. Microscopically, the characteristics of hydrothermal filling and alteration are remarkable. In gabbro, feldspar and dark minerals are metasomatized and the feldspar retains its original crystalline form, whereas the dark minerals have been micanized and epidotized. It can be seen that chlorite, epidote, mica, and other minerals are oriented, and the newly formed bright white hydrothermal quartz is filled between feldspar mineral grains (Figure 3a–c). Mica, chlorite, sericite, and quartz are fractured, and the altered min-

erals have shape-preferential orientation (Figure 3a–c). Minerals showing evidence of static recrystallization have pressure shadowing, and late hydrothermal quartz appears in the fractures of broken quartz grains (Figure 3c). These observations indicate that shear deformation leads to rock fragmentation and thermal alteration leads to mineral alteration and ductile deformation. In the Cu, Pb, and Zn metallogenic section, the quartz and alteration mineral fractures develop in the surrounding area and the new quartz cells fill the fractures with broken particles. The hydrothermal galena, sphalerite, pyrite, and other minerals fill the fractures of cataclastic minerals. These metal sulfides show no signs of fracture after shearing (Figure 3d). It is speculated that in the ore-forming hydrothermal fluids, gabbro intruded along the fracture and formed strong alteration, including the transformation of amphibole and biotite into chlorite and chlorite and the formation of a small amount of secondary quartz. Metallic minerals such as chalcopyrite, sphalerite, and galena are intact and associated with hydrothermal quartz. Some of the metallic crystals formed with the migration trace of hydrothermal filling and insufficient crystallization separation can be seen. All these phenomena indicate that the metallic minerals are transferred to the structure space through hydrothermal condensation crystallization.

Element Migration during Hydrothermal Activity

The migration of elements favors the aggregation of metallic elements during hydrothermal activity. The alteration of the host rock by the hydrothermal fluid resulted in a significant decrease in FeO, MgO, and CaO and a significant increase in $K_2O$, $Na_2O$, and $SiO_2$ in the gabbro of the present study. The distributions of REEs and trace elements in altered gabbro shows two patterns: some ore samples show patterns similar to gabbros, whereas others show different patterns. The altered gabbro show low total REE contents and depletion in Ce and Eu, as well as pronounced enrichment in Rb and depletion in Sr (Figure 5).

The fresh gabbros and hydrothermally altered gabbros from the Ping'an deposit have markedly different Pb, Zn, and Cu contents. The Pb and Zn contents in the hydrothermally altered gabbros are 10 to 100 times higher, and Cu is 10 to 100 times lesser in the fresh gabbros (Table 1). The Pb and Zn may have been supplied by the hydrothermal solution system; however, the Cu was extracted from gabbro, causing an uneven distribution of metal minerals within the deposit. In addition, Rb, Th, U, and Pb are relatively enriched, whereas Ba, Sr, Nb, and Ta are relatively depleted in the fresh gabbro. The high Fe/Mn and Zn/Fe values and trace-element characteristics indicate that the Ping'an Village gabbro was derived from the pyroxene-rich source of the subducted recirculated plates. The Nb/Ta and Th/Hf values are relatively stable in diabase and mineralized bodies. The Ba/Sr and Rb/Sr ratios are the highest and most variable in mineralized bodies. These characteristics indicate that the rock mass was strongly influenced by the later hydrothermal alteration, which resulted in the strong and weak fractionation of mobile and immobile elements, respectively. The hydrothermal activity led to the migration and exchange of some elements. Based on the above analysis, we propose that the source of Pb–Zn polymetallic minerals in the area of Ping'an Village was complex, with Pb, Zn, and Cu derived from the basic rocks and strata and Sn from the nearby Ping'an quartz rock mass. The fine cassiterite was extracted from the tin-bearing section by hydrothermal solution.

*5.4. Tin Deposits in the Northern Guangxi Province*

Cassiterite from tin deposits in northern Guangxi was formed in the Neoproterozoic stage, and the Neoproterozoic SN-trending fault zone controlled the crystallization of tin-containing hydrothermal solution precipitated from the acidic magma. This process formed the primary mineralized bodies, which were subsequently deformed (broken into traces) during the Caledonian shear process. For example, the Jiufeng and Nandaoao tin deposits are mainly granular and disseminated and fine (net) veins occur in the brittle and ductile shear zone of the Sibao Group of the Neoproterozoic and dolomite dikes. By comparing the regional deposits with the three typical deposits, the cassiterite particles

in Nandaoao and Jiufeng tin deposits are generally dissolved and broken, showing the traces of subsequent shear and hydrothermal alteration (Figure 13). The size of cassiterite in Nandaoao tin deposit is mainly between 100 and 150 μm, and the cassiterite is dissolved to different degrees, indicating unevenness. Part of the cassiterite does not have obvious zonal texture (Figure 13a). Part of the cassiterite in Jiufeng tin deposit has a complete zonal texture, but the fine-grained cassiterite, irregular cassiterite, and part of large cassiterite are without zones. Cassiterite minerals are mostly missing and rarely seen in full crystalline form. At the same time, the occurrence of widely exposed ultra-basic rock in these three deposits is obviously inconsistent with the ore body. For example, the tin ore-body in Jiufeng deposit is nearly SN-trending, while the diabase is EW or NW trending. The diagenetic age of metamorphic diabase is ~823 Ma, which is also significantly later than the crystallization age of cassiterite. Therefore, the ore-forming tin in the study area probably came from the Neoproterozoic magmatic rocks. Combined with the previous analysis, it can be determined that the cassiterite underwent a cataclastic process, followed by hydrothermal alteration for its dissolution. The tin–polymetallic deposits contain early (later deformed) cassiterite, influenced by late hydrothermal fluids containing Cu, Pb, Zn, and other metallic elements, which migrated through faults, joints, and fissures of host intrusive rocks. The deformation of the Caledonian orogenic period changed the mineral morphology of cassiterite and provided a space for the formation of the tin–polymetallic deposits. The Caledonian hydrothermal processes have altered the composition and spatial distribution of early ore bodies. The tin–polymetallic deposit in the northern Guangxi Province was probably formed due to early (Neoproterozoic) mineralization with late (Caledonian) base metal-bearing hydrothermal fluids.

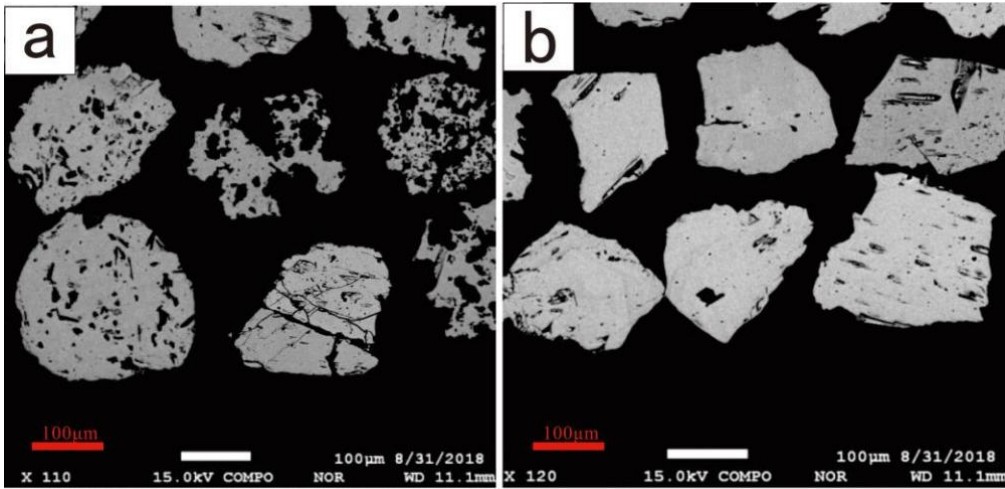

**Figure 13.** Cassiterite cathodoluminescence (CL) images from the tin deposit at the northern Guangxi Province. (**a**) CL images of the cassiterite in cataclastic quartz in Nandaoao; (**b**) CL images of the fragmented cassiterite in Jiufeng.

However, it should be pointed out that the Sn content of Ping'an village ore in the hand-held ore analysis instrument is up to 0.1% on average, but after ore analysis, it was found that the Sn content is only 10–20 times more concentrated than the stratum, which cannot constitute an ore body. From SEM analysis, we found some tiny cassiterite particles, which may have resulted from the far ancient crystallization (a tin-mining area in the south) and then formed, through the shearing action (shear some subtle cassiterite particles), cassiterite particles in the latter after hydrothermal handling, during migration to the ore-forming space. In other metal elements, crystallization occurred in the sealed edge of minerals. However, this conclusion needs to be confirmed or corrected by further research.

Based on the above analysis, we believe that the source of Pb–Zn polymetallic minerals in Ping'an village is more complex. The Pb, Zn, and Cu elements may be related to basic rocks and strata, and the Sn element is undoubtedly related to the evolution of the nearby

Ping'an quartz rock mass. The fine cassiterite has been extracted from the tin-bearing section using a hydrothermal solution during migration.

## 6. Conclusions

We obtained a zircon U–Pb age of 450.4 ± 6.7 Ma for the host gabbro, which confirms that Caledonian magmatism occurred in the northern Guangxi Province. The ages obtained may represent Caledonian gabbro intrusion events. The obtained age results provide a new basis for the final fusion of the Cathaysian and Yangtze plates after the Caledonian movement.

We consider that most of the polymetallic deposits in which cassiterites crystallized during the Neoproterozoic in the northern Guangxi Province, including the Ping'an deposit, were formed by early (Neoproterozoic) cassiterite crystallization and late (Caledonian) shear fracture, and later, ore-bearing hydrothermal alteration and superimposition.

**Author Contributions:** Conceptualization, Z.X. and S.L. (Shehong Li); methodology, Z.X.; software, Z.X.; validation, S.L. (Shehong Li), X.H. and J.W.; formal analysis, S.L. (Shehong Li); investigation, S.L. (Shehong Li); resources, S.L. (Shehong Li); data curation, Z.X.; writing—original draft preparation, Z.X.; writing—review and editing, Z.X.; visualization, Z.X.; supervision, S.L. (Saisai Li), Y.D. and C.P.; project administration, Y.D., C.P. and S.L. (Saisai Li); funding acquisition, S.L. (Shehong Li). All authors have read and agreed to the published version of the manuscript.

**Funding:** Guangxi Natural Science Foundation (2018GXNSFAA281260).

**Data Availability Statement:** The data that support the findings ofthis study are available on request fromthe corresponding author. The data arenot publicly available due to privacy orethical restrictions.

**Conflicts of Interest:** We declare that we do not have any commercial or associative interest that represents a conflict of interest in connection with the work submitted.

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
