# Peer review of "Zircon Dating, Geochemistry, and Metallogenic Significance of Early Paleozoic Mafic Rocks in Northern Guangxi Province, China"

_minerals, doi:10.3390/min12060672_

Round 1
Reviewer 1 Report
The manuscript deals with the ages of ore bodies in an area that is known for an extended geological history with rocks and mineralizationos of different ages. The data presented is interesting, but it could be better contextualized to make the reader understand their significance. Find below some considerations that I think need to be addressed to improve the manuscript and check the yellow outlines in the pdf for expressions or ideas that should be revised.
Introduction:
It is a bit confusing, as it mentions Sn polymetallic mineralizations and other base metal polymetallic mineralisations without describing them, nor mentioning their abundance or frequency in the region nor explaining the relationsihip between them. Are there other mineral deposits of Au or U or W nearby as in other parts of the province? They seem to have different ages, as well.
The objective of the study is not clear. The last paragraph points to an ‘integrated study of mineralization and host rock’ but the lines above say ‘dating combined with mineralogical geology based on previous studies’, so which is it? Integrated or based on previous studies? .
Figure 1d only show one ore body?
There are sentences with some English errors. I suggest to revise the section and enlarge it with more explanations to clearly state the main ideas and the objective of the manuscript.
Geological setting:
Add references in the first paragraph.
The photographs of Figure 2 are too small or lack quality to be able to distinguish the gabbro from the ore body. I recommend to add explanations of the metasomatisms and point the principal features in the photographs.
Section 2.2 is original for this manuscript? If this is so, this part could be integrated in the results section. If it comes from previous studies, they should be referenced. It is not clear to me how the ore body looks like: veins with sharp contacts with the gabbro? Or dissemination within the gabbro as well? Altered gabbro is considered ore body? Detailed pictures of field sections or hand specimens would help.
Results:
Section 4.1.2 title is ‘Altered gabbro’ but then the first sentence starts with ‘The ore contains…SiO2…’: clarify, please: what are the differences between the altered gabbro and the ore deposit? Is the altered gabbro a mineralized gabbro? Ore deposit is not the same as ore minerals.
Figure 3 would be easier to understand if the altered and fresh gabbros had different line colors.
’16 samples of ore’ in section 4.2: does this mean 16 samples of mineralized gabbros? If group I zircons is composed of only two zircons, how do you obtain two similar ages for them? Is this representative enough?
Try to separate results or obtained data from their interpretation: data interpretation should go to the Discussion section.
Discussion:
This section contains information that would be better placed either in the Introduction or in the Geological setting section. And there’s no need to repeat the results explained above.
Check the order and numeration of figures mentioned in the text.
The first paragraph of section 5.4 could be moved in the Introduction or Geological setting sections.
Has the idea that ‘The polymetallic tin deposits contain the early (and subsequently deformed) cassiterite, and were affected by later hydrothermal fluids containing Cu, Pb, Zn, and other metal elements, which were transported through faults, joints, and fissures in host intrusive rocks’ (pg 15) been published before in reference to other deposits of the area? It would be interesting to contrast this multiple ages of ores in different sectors.
The origin of base metals have not been much discussed.
Conclusions:
Is the sentence ‘We consider that most of the polymetallic deposits in which cassiterites crystallized during the Neoproterozoic in northern Guangxi, including the Ping’an deposit, were formed by early (Neoproterozoic) cassiterite crystallization and late (Caledonian) shear fracture and later ore-bearing hydrothermal alteration and superimposition. ‘ a new idea never been published before?
Reviewer 2 Report
This paper reported a dating result of zircon U-Pb age for gabbro that host Pb-Zn-Cu polymetallic veins and discussed its formation age and tectonic setting as well as the structure and compositional change due to later fluid metasomatism. My comments and suggestions are as follows.
1. The title is irrelevant to the major text. In this paper. the major text centers on the gabbro rocks, why does the title focus on the origin of a Pb-Zn-Cu deposit that is hosted by the gabbro. Actually, this paper presented little evidence on the origin of the Pb-Zn-Cu deposit. The current title is unacceptable.
2. Comparison between previous studies and this study. This study determined the main age as 450.4±6.7 Ma for the gabbro, however, previous studies have shown that those mafic and ultramafic rocks in the studied area formed during 860-740 Ma. Why such a large gap? A suggestion is that the authors should discuss the possibilities of dating result discrepancy.
3. Table 1 can not discriminate the data of altered gabbro from those of fresh gabbro.
4. The data of ore in Fig.6-Fig.11 were not shown in Table 1.
5. Section 5.3 is really weak. The timing of the hydrothermal fluid event? Microphotographic evidence?
6. Section 5.4 is irrelevant to this study, and I suggest deleting it.
7. I suggest the authors to enhance the image resolution for all the figures.
Round 2
Reviewer 1 Report
The revised manuscript is a big improvement with respect to the first version. For example, the new title makes a good change as it is more descriptive of what’s in the text, or the modifications in the Abstract that improve it. However, there are many aspects that I pointed in the revision that have not been yet addressed and all the manuscript contains many grammar or wording issues so that it should be revised by an English native speaker. Find below and in the attached manuscript other specific points to consider.
The Introduction section has been positively extended to include explanations about the Sn deposits and the objective is more clearly defined. However, the wording of some paragraphs is not correct and it difficults the comprehension of the geological history of the area. Figure 1d still shows only one ore body, correct the figure caption.
The Geological Setting section is better now but has not answered all may concerns. Explanations of section 2.2 should be moved to the Results section if the information it contains has not been published before and if it is your own collected data. If this is not the case and it is not new information, then it should be appropriately referenced and be kept in this section.
You mention in your Reply to the Review Report that ‘this is the first time for us to discover the mineralization in Caledonian rock in this area’. Does this mean that it has never been described before? If so, it is very important to publish a thorough charaxterisation (mineralogy, paragenesis, textures, dimensions...).
The addition of more pictures in Figure 2 is noted. It would be simpler to comprehend the mineralizations and their importance if the figure captions of the field photographs included more explicit descriptions of what’s important to be observed in each one.
The first aspect I’d expect of a new mineralization discovery in the Results section is the description of the ore bodies, tha is, dimensions, mineralogy, grain size, ore conentration, textures and paragenesis with illustrating photographs also under the microscope. It should be made clear in the text if the mineralization is limited to veins within an altered gabbro or consists of disseminated sulfides in the gabbro or both, as well as the grade or concentration of ore minerals in each part of the ore body. The last sentence of the Introduction reads ‘in this paper we present an integrated study of the geology, the structures, and mineralization of the Ping’an intrusive gabbro and its associated Pb-Zn-Cu deposit’, but until you present a good description this is not fulfilled.
Place all tables and figures in the text after being mentioned. For example, Table 1 should be positions within the Results section, not before. Some figures have been renumbered but other have remained out or order according to their citation within the text. Do not move the micro-photographs of ore mineral into the Discussion section as they illustrate the mineralization description. Table 4 has not been mentioned in the text.
The Discussion section could be improved with more up to the point explanations and comparisons.

Author Response
- The ore body characteristics you mentioned have been moved and redescribed.
-
The table position has been adjusted, and the table 4 has been deleted.
- English language and style have been required.
Reviewer 2 Report
The authors have made a major revision on this manuscript according to the first-round review comments. I have not comments any more.
Author Response
English spelling and sentences have been checked and revised.
Round 3
Reviewer 1 Report
The revised manuscript is a big improvement with respect to the first version. The English language has been mostly corrected.
However, there are many aspects that I pointed in the revision that have not been yet addressed.
The Geological Setting section still contains some description of the ore bodies, which could be eliminated now as it is repeated in the Results section.
I still do not understand what the ore body consists of (veins or dissemination?). The description of the ore bodies needs improvement. See the comments in the pdf.
The Discussion section could still be improved, see the comment in the pdf.
Find in the attached manuscript the specific points to consider.

Author Response
- Ore body descriptions in geological Settings have been removed.
- The description of ore bodies has been improved.
- It has been modified according to the PDF.